# Exploring the relationship between the food environment and preferences among schoolchildren in a low socioeconomic community in Sri Lanka: A GIS-based assessment

Chamil Priyanka Senevirathne[1]*, Prasad Katulanda[2], Padmal de Silva[3], Dilini Prashadika[4‡], Lalith Senarathne[1‡], Manoj Fernando[1‡]

**1** Department of Health Promotion, Faculty of Applied Sciences, Rajarata University of Sri Lanka, Anuradhapura, Sri Lanka, **2** Department of Clinical Medicine, Faculty of Medicine, University of Colombo, Colombo, Sri Lanka, **3** Department of Statistics, National Institute of Health Sciences, Kaluthara, Sri Lanka, **4** Department of Computing Sciences, Faculty of Applied Sciences, Rajarata University of Sri Lanka, Anuradhapura, Sri Lanka

☯ These authors contributed equally to this work.
‡ DP, LS and MF also contributed equally to this work.
* chamil@as.art.ac.lk

## Abstract

The food environment in school neighborhoods plays a crucial role in manipulating the food choices of schoolchildren. This study investigated the relationship between the food environment in neighborhoods and the dietary practices of government school students in a low socioeconomic setting in Sri Lanka. This cross-sectional study surveyed the neighborhood food environment of selected schools (n = 30) in the Monaragala District, Sri Lanka, using geographical information system (GIS) data and collected dietary information from a representative sample of schoolchildren (n = 603). Chi-square and Spearman correlation tests were performed using SPSS version 23.0 to estimate the associations between the food environment and BMI, while ArcGIS 10.4.1 was used to analyze the GIS data. The majority of the students (35.5%) were 15 years old, and approximately 51% were females. The mean BMI of the study participants was 18.14 (±3.28). More than 90% of outlets within proximity sold unhealthy foods. Consumption of confectionaries was 72.3% of the students, whereas healthy food choices ranged from 5% to 12%. A positive correlation between consuming unhealthy food and distance to outlets from school was observed (p<0.05). The risk of consuming low-nutrition food doubled (OR = 2.47, 95% CI: 1.52–3.89) among the students studying in schools where a larger proportion of energy-dense food was sold in closer proximity. In conclusion, the density and proximity of outlets that sell food with low nutrients in the school neighborhood environment were positively associated with students' unhealthy food item choices.

**Data Availability Statement:** All relevant data are within the manuscript and its Supporting information files.

**Funding:** The author(s) received no specific funding for this work.

**Competing interests:** Authors have declared that no competing interests exist.

## Introduction

The neighborhood food environment in the school is defined as the physical infrastructure and overall surroundings adjacent to school premises where food is both sold and consumed. This encompasses the spectrum of food items available in these locations, considering not only their availability but also the nutritional composition they offer [1]. Food markets, retail shops, street food stalls, boutiques, tea shops, and various other establishments catering to people's dietary needs collectively constitute the primary components of a food environment. Moreover, these environments exhibit diversity based on specific contexts and detailed settings, such as the home food environment or the school food environment [1]. Nevertheless, the decision to consume food from these food destinations is influenced by several factors, including the type of food accessible, the convenience of obtaining food, and the price range of the available food.

The presence of unhealthy food options near schools significantly shapes the dietary choices of schoolchildren, particularly during adolescence. This phase of life is pivotal for the development of both cognitive and physiological aspects that impact an individual's overall well-being across the lifespan [2, 3]. A positive lifestyle and behavioral practices during this period reduce the risk of chronic diseases in early adulthood [4]. Marketing strategies of the food industry influence schoolchildren to reach food destinations such as fast food huts, grocery outlets, and cafeterias to fulfill their day-to-day dietary requirements [5]. A positive association between the availability of food outlets nearby and eating habits and the nutritional status of people across different settings and populations has already been established [6, 7]. A recent study demonstrated that neighborhood food destinations provide a significant amount of calories to school children in the USA [8]. However, the presence of energy-dense foods promoted through child-focused marketing substantially influences the food choices of schoolchildren [9]. Several studies have shown that children and adolescents who have easy access to unhealthy food outlets near their schools tend to have greater consumption of fast food, sugary beverages, and snacks. This is often associated with an increased risk of obesity and related health issues [10–13]. The presence of healthy food, on the other hand, has also been linked to better dietary choices. Therefore, the school neighborhood food environment plays a crucial role in influencing eating patterns, food preferences, and further schoolchildren, as they are exposed to a wide range of unhealthy food and beverage items during school hours.

Sri Lanka is a middle-income country with diverse socioeconomic backgrounds across different communities. Approximately 20% of the population comprises school-age children and adolescents [14]. The majority (57.3%) of schoolchildren are provided free education through 10,165 government schools, while others attend private (37.2%) aided (8.5%) and other schools (3.3%) [15] and all provincial schools. The Sri Lankan government introduced a school canteen policy in 2006 to ensure access to nutritional food within school premises. This policy is focused mainly on changing the food environment within school premises. Although having a policy to regulate food availability on school premises is vital for improving the nutritional practices of children, school neighborhoods should also be taken into consideration when changing the school food environment. Comprehending the food environment, encompassing the nutritional value of accessible food items both within and surrounding neighborhoods, significantly affects the nutritional well-being of schoolchildren. While a handful of studies have probed the school nutritional environment and its correlated factors in urban settings in Sri Lanka [16], there is a notable dearth of evidence concerning economically disadvantaged communities. Consequently, there is a discernible gap in the body of evidence on the school food environment in Sri Lanka. Addressing this gap holds paramount importance in generating the essential additional evidence required for the efficacious implementation of food environment

policies aimed at schools. The objective of this study was to explore the association between the neighborhood food environment and the dietary practices of schoolchildren within a low socioeconomic community in Sri Lanka. This exploration was conducted through the lens of geographical accessibility, utilizing a geographic information system (GIS) as a fundamental tool.

## Methodology

### Study design

A cross-sectional study was conducted in government schools under the Provincial Department of Education in the Moneragala Education Zone of Moneragala District of Sri Lanka, which has been identified as one of the economically deprived districts in Sri Lanka. Poverty indicators in Sri Lanka [17] were used to define the economic characteristics of the study setting.

### Study population

**Primary target population.** Schoolchildren aged eight to eleven (aged 13 to 16 years) from government schools in the Moneragala education division in the Uva province of Sri Lanka were selected as the study population. Students with cognitive disabilities and living with chronic diseases requiring long-term drug treatment and follow-up were excluded from the study.

**Secondary target population.** Individuals who sold ready-to-eat food items, processed food, confectionaries, or other types of snacks were considered for the data collection. All the outlets that sold food items (except for mobile carts and mobile food sellers) located 300 meters from the school border were selected for the assessment. Other outlets that did not sell any type of food were excluded from the survey.

### Sample size calculation

In low socioeconomic groups, the proportion of students who access neighborhood food outlets was considered for the sample size calculation. Since there were no previous studies reporting the prevalence of this disease, the sample size was calculated to determine at least 50% of the population. The sample size was calculated using the formula for cross-sectional studies:

$$N = z^2 p(1 - p)/d^2$$

[18].

Together with the hypothesized prevalence of accessing neighborhood food outlets among schoolchildren (50%), a 0.05 level of significance (precision d = 5%), Z = 1.96 standard normal deviation at 95% CI was included in the calculation.

Previous studies suggested including the designed effect 1.5 to ensure the statistical validity of the sample selection procedure. To minimize the error due to clustering, the calculated sample size was multiplied by the design effect (D), which was taken as 1. 516 [19]. The sample size was 576; to compensate for the 10% nonresponse rate, another 58 participants were added to the sample, for a final sample size of 630. Multistage stratified cluster sampling was used to select the required number of clusters (n = 30). To describe the selection of the secondary sample, all the outlets that sold food within a 300-meter radius of the school's main gate were considered for the data collection.

## Data collection procedure and study instruments

The data collection procedure started on 12[th] November 2016, whereas the data collection process ended on 22[nd] June 2018. The validated version of the Global School Health Survey (GSHS) questionnaire was used to collect demographic and dietary information from the participants. This questionnaire was validated by the Ministry of Health Sri Lanka to collect school nutrition data in Sri Lanka as a part of global research conducted by the World Health Organization in 2016 [20]. This interviewer-administered questionnaire collected two types of information. The first part investigated sociodemographic data, including age, sex, parental education, and family income, while the second section gathered data on the dietary habits of school children, focusing on their consumption patterns of nine food markers (five unhealthy food markers and four healthy food markers) during the previous 30 days from shops located in school neighborhoods. For this study, five unhealthy food markers were selected: starch food, carbonated drinks, confectionaries, short eats, and fried food, and the healthy markers were fruits, pulses, dairy products, and healthy beverages. The questionnaire was translated into Sinhala and Tamil languages and then back-translated to English to ensure the accuracy of the content. The same one was pretested with a sample to observe the understandability and accuracy of the responses.

Since recommendations for the availability of healthy and unhealthy food markers in the school neighbourhood environment are absent, food availability in the school vicinity was assessed based on the recommendations of the School Canteen Policy in Sri Lanka [21]. The School Canteen Policy has defined designated health food markers that are recommended for availability on school premises. For instance, the following food items, rice, vegetables, fruits, fish, eggs, sandwiches, milk, tea, fruit drinks, and porridge, have been introduced as healthy markers, while food items containing high sugar (confectionaries, biscuits, chocolates, cakes), high salt (processed food, bites, mixtures), and high-fat food (sweetmeats, deep-fried food) are prohibited from being sold on school premises. An observation checklist, which was developed based on the recommendations of the school canteen policy, was utilized to determine the availability of different food markers in school proximity. The School Canteen Policy Each shop was visited by the investigator to identify the availability of healthy and unhealthy food markers, and the information was recorded under the selected categories of the checklist. Healthy and unhealthy food markers were defined according to the recommendations of the school canteen policy [21]. The food environment in the neighborhood, as considered in the present study, includes all establishments selling food markers within a 300-meter radius of the school. A preliminary geographical analysis was conducted by the investigators to establish the appropriate buffer distance. of the present study was considered all outlets selling food within 300 metres of the school. This identification was performed based on a sample geographical analysis performed by the investigator before determining the buffer distance. To conduct this sample analysis, two schools were randomly selected from each division, and shops within 500 meters of these schools were identified. Subsequently, shopkeepers were interviewed to ascertain the frequency of school student visits before and after school hours. This investigation revealed that a majority of the students resided within walking distance of their schools, and the average maximum distance to shops in the school vicinity was approximately 300 meters. Consequently, a 300-meter radius was chosen as a parameter for the data collection. The 300-meter distance was subsequently subdivided into three distinct buffer zones to determine the impact of neighborhood environments on the dietary preferences of schoolchildren. To comprehensively understand the food context of the outlets within these zones, the investigators employed a Garmin ETrex 10 GPS device to capture the coordinates of both the outlets and schools. For the collection of school coordinates, the procedure involved

standing directly in front of the school gate and capturing the school coordinates under a clear sky. Following this, the investigator walked to each shop within each buffer zone, gathering coordinates while positioned in front of the respective shops, once again ensuring clear sky conditions. The entire process, including the data, time, and coordinates, was meticulously recorded manually for future analysis and then tabulated in Microsoft Excel 2010. To establish physical measurements of the study participants, height and weight were measured to the nearest 0.1 cm and 0.1 kg, respectively, by trained data collectors. Standardized procedures for anthropometric measurements were used to obtain both measurements [22].

### Data analysis

During the analysis phase, descriptive methods were employed in alignment with the study objective. Proportions were calculated as percentages of the categorical variables. The examination of relationships between shop proximity and density within 100-meter, 200-meter, and 300-meter buffer zones. A chi-square test was performed to determine the association between dependent variables (consumption of unhealthy food makers: starch-food, short-eats, confectionaries, etc.) and independent variables (proximity and density of food-selling establishments). In addition, multivariate logistic regression was performed to determine the level of predictors (establishments selling food items located in three-level buffer zones (100 metres, 200 metres, and 300 metres)) of consuming unhealthy food makers: starch-food, short-eats, confectionaries, etc. Statistical significance was assessed at the (95%, CI) $p < 0.05$ level. The Statistical Package for the Social Sciences (SPSS 23.0) was used to analyse the data.

To analyse the GIS data, ArcGIS 10.4.1 software was used in the present study to analyse spatial neighbourhood information. The GPS coordinates of outlets and schools were used to create a point layer on ArcMap, and these two layers were used to analyse the distance from each school to the outlets. Every school location created multiple buffer zones, and these buffer zones clearly showed the surrounding food outlets of the school environment. Shop density and proximity were defined as the operational variables for the number of outlets located within given buffer zones and distance from the school, respectively.

### Ethics statement

The study took necessary steps to ensure the ethical validity of the study. All the participants were provided with an information sheet about the study procedure, and written consent was obtained after providing an opportunity to ask questions. In addition, consent was obtained from the parents and guardians of the schoolchildren. Steps were taken to maintain the confidentiality of the data and the privacy of the study participants by collecting anonymized information. The study protocol was reviewed and approved by the Ethical Review Committee of the Faculty of Medicine, University of Colombo; **Approval number EC-14-105**.

### Results

With a response rate of 95.7%, 603 students were recruited from 30 schools to ensure representation across all three education zones within the Moneragala District in Sri Lanka. Among the selected sample, a predominant proportion of males (47.7%) and females (52.3%) were 15 years old, while notably, more than half (56.4%) of the enrolled students lived in low-income families. The mean BMI for males was 17.6 kgm$^{-2}$ (±3.4), while for females, it was 18.5 kgm$^{-2}$ (±3.1). A large proportion of males (74.7%) and females (57.1%) were underweight (Table 1).

Table 2 displays the distribution of food outlets across the individual buffer zones. The analysis incorporated a total of 97 outlets situated within a 300-meter radius. Notably, a significant

**Table 1. Sociodemographic characteristics and weight status of the students included in the Moneragala education zone (n = 603).**

| Characteristics of study participants | Frequencies | |
|---|---|---|
| | **Male (%)** | **Female (%)** |
| Gender | 292 (48.4) | 311 (51.6) |
| Age group | | |
| 13 years | 40 (42.1) | 55 (57.9) |
| 14 years | 97 (51.6) | 91 (48.4) |
| 15 years | 102 (47.7) | 112 (52.3) |
| 16 years | 54 (50.9) | 52 (49.1) |
| Mother's education level | | |
| No education | 58 (51.7) | 54 (48.3) |
| Primary education | 122 (49.5) | 124 (50.5) |
| Secondary education | 113 (46.1) | 132 (53.9) |
| Father's education level | | |
| No education | 72 (45.5) | 86 (54.5) |
| Primary education | 166 (57.5) | 123 (42.5) |
| Secondary education | 86 (50.9) | 83 (49.1) |
| Monthly income | | |
| Low | 186 (54.7) | 154 (45.3) |
| Middle | 72 (45.0) | 88 (55.0) |
| High | 44 (48.2) | 59 (57.2) |
| Weigh status of the study participants | | |
| Underweight | 219 (74.7) | 177 (57.1) |
| Normal weight | 56 (19.1) | 101 (32.5) |
| Overweight | 18 (5.2) | 32 (10.4) |

Table 1: illustrates the demographic characteristics and weight status of the study participants. *The income categories were defined as follows: low (<10000 LKR, middle (10000–30000 LKR), and high (>40000 LKR).

proportion of these outlets were concentrated within a 100-meter buffer zone, with a comparatively lower count observed within a broader 300-meter buffer zone.

A graphic representation of the school distribution among the three education zones is shown in Fig 1. The distribution of schools and shops in zone one, the Moneragala Education Zone, is shown on map one. The distribution of schools and shops in zone two, the Biliga Education Zone, is shown on map two. The school neighborhood environment in zone three, the Wellawaya Education Zone, is explained on map three. To illustrate how the three buffer zones that around each school are distributed, examples are provided for each zone.

**Table 2. Shop distribution in distinct buffer zones in each educational zone, Moneragala (n = 30).**

| Education divisions | Number of schools | Number of food outlets | | | |
|---|---|---|---|---|---|
| | | Buffer zone 1 (100 meters) | Buffer zone 2 (200 meters) | Buffer zone 3 (300 meters) | Total |
| Moneragala | 10 | 15 (50%) | 09 (30%) | 06 (20%) | 30 (100%) |
| Biblia | 10 | 12 (44%) | 08 (29%) | 07 (27%) | 27 (100%) |
| Wellawaya | 10 | 23 (58%) | 08 (20%) | 09 (22%) | 40 (100%) |
| Total | 30 | 50 (52%) | 25 (26%) | 22 (22%) | 97 (100%) |

Table 2: displays the distribution of shops within the 1st, 2nd, and 3rd buffer zones across three educational divisions in the Moneragala Education Zone.

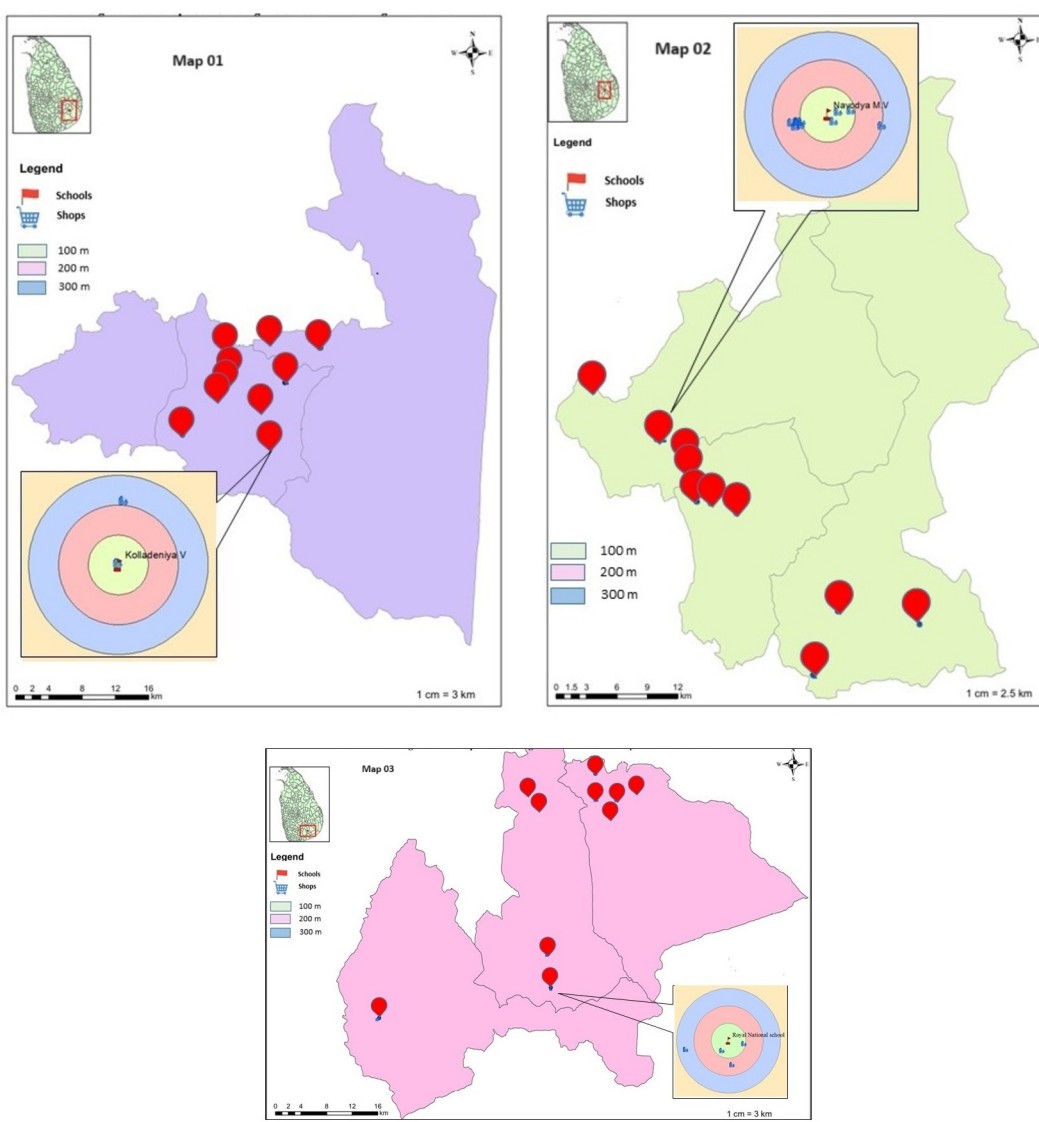

**Fig 1. Distribution of schools and neighborhood food shops within 300-meter buffer zones in three education zones.** The location of schools in the Moneragala education zone is depicted in Map one, while the school food environments in the Bibila and Wellawaya education zones are shown in Maps two and three.

Fig 2 outlines the proportions of healthy and unhealthy food items consumed by schoolchildren. The data revealed that confectionaries and short eats were the preferred choices, accounting for 72.0% and 75.0%, respectively, of consumption. On the other hand, starch-based foods, such as bakery items, were the least common unhealthy food items consumed (32.0%) among the study participants. In addition, the consumption of healthy food markers from food establishments in the school vicinity was observed to be less than 10%.

Table 3 describes the percentages of establishments selling unhealthy food markers, including starch food, short eats, and confectionaries, which were found to be 92.8%, 92.8%, and 100.0%, respectively. Furthermore, 82.5% of the shops in the school vicinity offered carbonated drinks. In addition, no establishments provided healthy food items such as pulses or nutritious beverages, while only 12.3% of the shops offered fruits for sale.

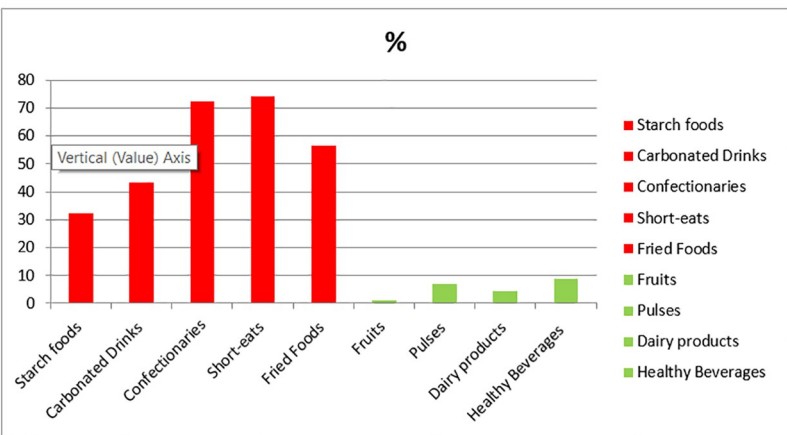

**Fig 2. Percentages of healthy and unhealthy food makers consumed by the study participants.** Unhealthy food markers adhered to the recommendations of the school canteen policy, Sri Lanka. starch food (bakery items, biscuits, rotties, hoppers, etc.); carbonized drink (cola, fizzy drinks); confectionaries (toffee, chocolates, iced packets); short-eats (rolls, cutlets, patis, and other pastries): fried food (bites and other deep fried food items).

Table 4 explains the association between the prevalence of food establishments offering unhealthy food within a 100-meter radius and the associated odds of consuming such food markers. Compared to those of schools with a minimal number of food establishments, the odds of eating deep-fried food were estimated to be 1.41 (95%, CI = 1.04–2.15), and the possibility of consuming confectionaries increased by 1.7 times (95%, CI = 0.83–3.95). Furthermore, significant associations were identified concerning the consumption of confectionaries (OR = 1.53, 95% CI = 1.09–2). 34) and carbonated drinks (OR = 1.4, 95%, CI = 1.04–2.15) in schools located within 200 metres of food establishments (four or more) compared to those with minimal (none or three) or three such establishments.

By adjusting for factors such as gender, income, and parental education through multivariate logistic regression analysis, a noteworthy association was indicated between proximity and

**Table 3. Classification of food establishments selling healthy and unhealthy food markers (N = 97).**

| Food items | Number of outlets | % |
|---|---|---|
| *Unhealthy food markers* | | |
| Starched-food | 90 | 92.8 |
| Short-eats | 90 | 92.8 |
| Confectionaries | 97 | 100 |
| Fried food | 97 | 98.9 |
| Carbonated beverages | 80 | 82.5 |
| *Healthy food markers* | | |
| Pulses | 00 | 00 |
| Fruits | 12 | 12.3 |
| Dairy products (yogurt/curd) | 68 | 70.1 |
| Healthy beverages (Herbal drinks/fresh milk/fruit juice) | 00 | 00 |

Table 3 presents the distribution of healthy and unhealthy food markers among the food establishments situated in the school vicinity. These food markers adhere to the criteria outlined by the recommendation of the school canteen policy in Sri Lanka.

**Table 4. Bivariate analysis of the associations between the consumption of unhealthy food markers among study participants and the density and proximity of establishments (n = 603).**

| Consumption of unhealthy food markers (Yes Vs No) | 1st buffer zone (4 or more establishments that sell unhealthy food makers) | | | 2nd buffer zone (4 or more establishments that sell unhealthy food makers) | | | 3rd buffer zone (4 or more establishments that sell unhealthy food makers) | | |
|---|---|---|---|---|---|---|---|---|---|
| | r | OR | P | r | OR | P | r | OR | P |
| Deep-fried food | 0.344 | 1.41 | 0.031 | -0.010 | 1.036 | 0.803 | -0.012 | 1.041 | 0.771 |
| Short-eats | 0.042 | 1.331 | 0.046 | 0.013 | 1.043 | 0.745 | -0.023 | 1.069 | 0.568 |
| Confectionaries | 0.202 | 1.745 | 0.006 | 0.173 | 1.513 | 0.031 | 0.031 | 0.895 | 0.450 |
| Bakery products | 0.021 | 1.047 | 0.603 | -0.018 | 1.091 | 0.660 | 0.022 | 1.160 | 0.584 |
| Carbonated drinks | 0.112 | 1.287 | 0.047 | 0.066 | 1.413 | 0.033 | 0.032 | 0.898 | 0.434 |

Table 4: displays the results of the chi-square analysis examining the association between the dependent variable (consumption of unhealthy food items (yes) and no) and the independent variable (density of establishments selling unhealthy food items). *

dietary choices among schoolchildren. The findings suggested that the availability of unhealthy food markers within 100 metres of school premises is associated with a 3.2-fold greater likelihood of consuming deep-fried food (95% CI 1.612–7.643), a 1.5-fold greater likelihood of consuming short eats (95%, CI 0.814–2.347), and a 2.0-fold greater probability (95% CI 1.011–4.689) of consuming confectionaries. In addition, students attending schools with unhealthy food selling outlets within a 200-meter buffer zone are likely to increase their consumption of deep-fried food and carbonated drink by 2.424 times (95% CI, 1.781–4.212) and 2.15 (95%, CI: 1.028–3.986) times, respectively, compared to schools without such food outlets in the vicinity. However, our analysis did not reveal a significant association between the establishments located in the third buffer zone (300 metres from school) and the consumption of unhealthy food markers among the study participants (Table 5).

## Discussion

Our study revealed that nearly all food outlets situated within a 300-meter proximity were engaged in selling unhealthy food, whereas only a limited number of outlets adhered to the healthy item criteria defined by the school canteen policy in Sri Lanka. On the other hand, this study revealed that a larger proportion of children (58.1%) consumed food from outlets located in the school neighbourhood environment and from discretionary snacks, and most types of food consumed by them were unhealthy. Moreover, this study indicated that the food environment of school settings has a significant relationship with the dietary choices of children in those schools. This relationship is reinforced as the density and proximity of food outlets in school neighborhoods increase.

In this study, a greater percentage of unhealthy food availability was observed in school proximity. These findings are in line with previous studies reporting a greater availability of unhealthy food items in the school environment [23, 24]. In agreement with the literature [25, 26], our study revealed that males consumed more food from the nearest food outlets. Furthermore, sugary beverages, bakery items, and deep-fried foods were the most common food items consumed, whereas less demand for healthy food options was observed among schoolchildren. Although the present study did not explore the reasons for the greater availability of unhealthy food in the school environment, a recent study further suggested that students may be culturally influenced by consuming such energy-dense foods [27]. On the other hand, recent studies have demonstrated that higher levels of fast food consumption among schoolchildren may be due to the availability of such food items in the school environment [26–28]. Hence, it is

**Table 5. Association between the presence of establishments selling unhealthy food in local school surroundings and the consumption of unhealthy food markers by schoolchildren (n = 603).**

| Consumption of Food markers (Yes Vs No) | The presence of establishments selling unhealthy markers | | | | | |
|---|---|---|---|---|---|---|
| | | | | | 95% CI | |
| | B | S.E | Adjusted (OR) | P | Lower | Upper |
| *First buffer zone (100 meters)* | | | | | | |
| Deep-fried food | 1.176 | 0.229 | 3.246 | 0.000 | 1.612 | 7.643 |
| Short-eats | 0.421 | 0.199 | 1.534 | 0.034 | 0.814 | 2.347 |
| Confectionaries | 1.398 | 0.244 | 2.048 | 0.000 | 1.011 | 4.689 |
| Bakery products | -0.709 | 0.444 | 0.492 | 0.111 | 0.206 | 1.176 |
| Carbonated drinks | 1.176 | 0.229 | 2.240 | 0.000 | 1.134 | 4.896 |
| *Second buffer zone (200 meters)* | | | | | | |
| Deep-fried food | 1.231 | 0.240 | 2.424 | 0.000 | 1.781 | 4.212 |
| Short-eats | -0.250 | 0.126 | 1.779 | 0.047 | 1.069 | 2.999 |
| Confectionaries | 0.123 | 0.135 | 1.131 | 0.361 | 0.869 | 1.473 |
| Bakery products | 0.043 | 0.100 | 1.044 | 0.688 | 0.858 | 1.270 |
| Carbonated drinks | 1.049 | 0.243 | 2.155 | 0.000 | 1.028 | 3.896 |
| *Third buffer zone (300 meters)* | | | | | | |
| Deep-fried food | -0.011 | 0.070 | 0.989 | 0.874 | 0.864 | 1.134 |
| Short-eats | 0.029 | 0.062 | 1.029 | 0.644 | 0.911 | 1.163 |
| Confectionaries | 0.087 | 0.089 | 1.091 | 0.328 | 0.917 | 1.298 |
| Bakery products | -0.158 | 0.073 | 1.854 | 0.032 | 1.179 | 3.985 |
| Carbonated drinks | 0.047 | 0.076 | 1.048 | 0.534 | 0.934 | 1.216 |

Table 5 presents the results of the multivariate logistic regression analysis, where adjustments were made for factors including age, sex, income, and parents' education level. The analysis was conducted with a 95% confidence interval (CI), and statistical sign

possible to argue that the increasing availability of unhealthy food increases the demand for such food by schoolchildren. Our study reaffirmed this fact, as greater consumption of energy-dense food from closer proximity was observed among participants.

This study revealed a significant correlation between opting for unhealthy food and the proximity of unhealthy food options within a 100-meter radius of the school premises. This association is in line with the findings of previous studies conducted in the United Kingdom [29] and South Korea [30]. These findings underscore the powerful influence of the local food environment density and immediate availability on the dietary preferences exhibited by students. Such insight holds the potential to inform targeted interventions aimed at promoting healthier food options within proximity to schools. Although the present study involved schoolchildren from low- and middle-income countries, the findings are comparable with those of studies from high-income countries [31]. Therefore, this study re-emphasizes the need to consider evidence from settings with varied socioeconomic statuses when developing policies on the impact of the school food environment.

Existing evidence indicates that the solidity of food outlets coupled with the availability of unhealthy food options influences the dietary choices of schoolchildren [32]. In contrast, this study showed that the availability of unhealthy food markers within 100-meter and 200-meter buffer zones increased the risk of consuming energy-dense food, such as confectionaries and carbonated beverages, by schoolchildren. These findings are in agreement with those of a previous study that established an interaction effect between the availability of unhealthy food in walk-shed food outlets and the dietary behavior of school-going adolescents [33]. Consistent

with recent findings, this study indicated that the outlet density in the first and second buffer zones was positively associated with the food consumption of the students [31–33]. Hence, the present study re-emphasizes that proximity can be attributed to an increase in the unhealthy food choices of school children. However, several studies have found no relationship between school environment and the consumption of energy-dense food by school children [5, 34, 35]. This disparity may be attributed to the fact that most of the studies being compared had framed urban communities in developed countries and assessed mostly the supermarkets, convenience stores and fast food restaurants where students may not have access to food. On the other hand, this study examined only food destinations located within walking distance, whereas most of the studies from high-income settings considered comparatively extended buffer zones for the analysis based on available computing facilities. The findings of our study demonstrated the importance of enhancing current nutrition-related policies through the incorporation of regional factors, which encompass demographic, socioeconomic, and cultural factors as well as individual characteristics. This approach ensures the equitable distribution of policy benefits across the entire country.

In this study, a substantial proportion of the participants were underweight, and it can be concluded that consuming energy-dense food is important for attaining their ideal weight. Nevertheless, empirical evidence demonstrates that the consumption of nutrient-deficient food serves as a pivotal factor in the nutritional shift from underweight to overweight or obese [36]. In addition, recent evidence affirms that exposure to junk food options during childhood severely influences individuals' dietary culture, food preference, and attitudes toward unhealthy dietary practices [37]. Furthermore, frequent fast food consumption is likely to decrease the odds of sufficient dietary habits such as lower intake of fruits and vegetables [37, 38]. Our study confirms this reality by highlighting a reduced inclination toward healthier food options, as advised by the school canteen policy in Sri Lanka.

The key strength of our study lies in its pioneering use of the geographical information system (GIS) to offer a comprehensive evaluation of how the school neighbourhood food environment influences the dietary preferences of schoolchildren in rural Sri Lanka. In addition, this study introduced a mechanism to define, characterize, and quantify the school food environment through the application of a geographical information system. Moreover, the study investigated the influence of the proximity and density of food outlets on the nutritional practices of schoolchildren in low-income communities. Therefore, our findings address the knowledge gap regarding the impact of the school food environment on the eating habits of schoolchildren in low-income communities in Sri Lanka.

One of the limitations is that this study examined all food outlets within the defined buffer zone, in contrast to many other studies that classified these food-selling establishments based on criteria such as range of food choices, geographical placement, and consumer-oriented amenities [39, 40]. Consequently, this approach potentially constrains our ability to fully explore how the food preferences of schoolchildren are influenced by the physical attributes of these food outlets. The present study also did not consider factors such as the purchasing power of students, affordability of food, quantity of snacks, or other motivations for consuming food from neighborhood outlets. Another limitation of the study pertains to the method used in defining the food preferences of schoolchildren, which relies on assessing the frequency of consumption of unhealthy food items without considering quantitative analysis of food consumption patterns. Therefore, future studies should employ methods such as 24-hour dietary recall and food frequency questionnaires to provide a more detailed understanding of the dietary habits and preferences of the study population.

## Conclusion

The present study investigated the dynamics of food availability within a defined buffer zone surrounding the school environment by examining the variety of food markers present in the immediate vicinity. Furthermore, this study aimed to establish an association between the density and proximity of food outlets and the consumption of unhealthy food makers from establishments located in school neighbourhood environments. The results emphasized a concerning trend: a pronounced prevalence of unhealthy food choices within the food outlets adjacent to schools. Strikingly, more than half of the surveyed population opted to obtain unhealthy food from nearby school neighbourhood food establishments. The study findings also revealed a notable preference for energy-dense food items among the study participants. In essence, this study emphasized the substantial influence of the characteristics of the neighbourhood food environment, specifically, neighbourhood density and proximity.

## Recommendations

The findings of the present study highlight that a healthy food environment around school premises is important for ensuring the nutritional behavior of schoolchildren in low-income communities. Therefore, measures to create and sustain a healthy food environment should be taken at the school level and at the policy level. The school canteen policy, which is the only available policy for schools, covers merely the food environment within school premises in Sri Lanka. However, this study does not address the influence of crucial neighborhood food environments, as highlighted by this study. Hence, in Sri Lanka, there is a notable policy gap that needs to be addressed to ensure the nutritional value of the school neighborhood environment. Therefore, expanding existing policies regarding the school food environment beyond school premises is recommended to include school neighborhoods. Such initiatives can be based on similar existing policies to address other public health issues related to adolescents; the National Alcohol and Tobacco Act (NATA) [41] prevents the sale of cigarettes within the premises and proximity of schools. The findings of the present study unequivocally illustrate the profound impact of the school environment on nutritional practices, shaping the food preferences of schoolchildren. Administrative bodies, notably the Ministry of Education and Ministry of Health in Sri Lanka, stand to derive substantial benefit from the primary conclusion of this study. These insights can serve as a foundation for bridging the policy gap by helping individuals focus on minimizing the accessibility of unhealthy food while concurrently promoting healthy options within the school neighborhood environment. In addition to establishing policies and mechanisms to monitor policy, encouraging health promotion initiatives for behavioural changes among students may contribute to addressing the issues revealed in this study.

## Supporting information

**S1 File.**
(PDF)

**S1 Data.**
(SAV)

## Acknowledgments

The authors of the study express their heartfelt gratitude to all participants of the study, including school children and shopkeepers, whose enthusiastic participation was invaluable. Special

recognition is extended to school staff for generously granting permission to access the school premises for data collection. Finally, our sincere appreciation goes to the administrative body of the Department of Education, Uva Province, Sri Lanka, for their kind authorization, which enabled the study to be conducted across three divisions.

## Author Contributions

**Conceptualization:** Chamil Priyanka Senevirathne.

**Formal analysis:** Dilini Prashadika.

**Methodology:** Chamil Priyanka Senevirathne.

**Software:** Dilini Prashadika.

**Supervision:** Prasad Katulanda, Padmal de Silva.

**Writing – original draft:** Chamil Priyanka Senevirathne.

**Writing – review & editing:** Dilini Prashadika, Lalith Senarathne, Manoj Fernando.

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
