## [Decision Letter · Decision Letter 0]

19 Jan 2024

PONE-D-23-28601Exploring the relationship between food environment and preferences of school children in low a socioeconomic community in Sri Lanka ; A GIS based assessmentPLOS ONE

Dear Dr. Senevirathne,

Thank you for submitting your manuscript to PLOS ONE. After careful consideration, we feel that it has merit but does not fully meet PLOS ONE’s publication criteria as it currently stands. Therefore, we invite you to submit a revised version of the manuscript that addresses the points raised during the review process.

We look forward to receiving your revised manuscript.

Kind regards,

Larissa Loures Mendes, Ph.D.

Academic Editor

PLOS ONE

Journal Requirements:

2. In the online submission form, you indicated that [Data is securely maintained with stringent measures in safeguard its confidentiality. Nevertheless, the author is ready available to provide the data upon request any time.]. 

3. We note that Figure 1 in your submission contain map/satellite images which may be copyrighted. All PLOS content is published under the Creative Commons Attribution License (CC BY 4.0), which means that the manuscript, images, and Supporting Information files will be freely available online, and any third party is permitted to access, download, copy, distribute, and use these materials in any way, even commercially, with proper attribution. For these reasons, we cannot publish previously copyrighted maps or satellite images created using proprietary data, such as Google software (Google Maps, Street View, and Earth). For more information, see our copyright guidelines: http://journals.plos.org/plosone/s/licenses-and-copyright.

Additional Editor Comments:

Dear Authors,

Thank you for the opportunity to review the manuscript. After the reviewers' evaluation, I suggest that the suggestions be inserted into the text and the reviewers' questions answered.

Yours sincerely

Reviewers' comments:

Reviewer's Responses to Questions

**Comments to the Author**

1. Is the manuscript technically sound, and do the data support the conclusions?

Reviewer #1: Yes

Reviewer #2: Partly

2. Has the statistical analysis been performed appropriately and rigorously? 

Reviewer #1: Yes

Reviewer #2: No

3. Have the authors made all data underlying the findings in their manuscript fully available?

Reviewer #1: Yes

Reviewer #2: Yes

4. Is the manuscript presented in an intelligible fashion and written in standard English?

Reviewer #1: Yes

Reviewer #2: No

5. Review Comments to the Author

Reviewer #1: In the introduction, the authors seek to explain the dimensions of the FAO's definition of school food environment, however they do not address food prices. I recommend going through this dimension, even if the focus is on availability, as it was discussed about the nutritional content of foods and marketing.

The content present at the end of line 89 and line 90 should be included only in the methodology.

The maps created in figure 1 are ineligible. It is essential to improve the resolution of the images.

The study is relevant and innovative for the country in which it was carried out (low-income country), this study topic is important in low- and middle-income countries. It would be interesting to point out at the end of the manuscript (before limitations) the importance of future research with the aim of understanding the information environment in and around these schools and also evaluating the presence of informal food trade. These are important points considering the surrounding context, especially considering low-income countries (the presence of informal commerce is very common, students possibly frequent these businesses).

Reviewer #2: Overall impression

I think this study offers a meaningful contribution to the literature on the school food environment, particularly as it supplies primary individual-level data regarding students and the food environment in the school neighborhood. Additionally, studies in this field are limited in low- and middle-income countries. However, after reading the text, I have several uncertainties that require clarification. I believe substantial modifications are necessary for the manuscript to meet publication standards. Furthermore, though it's not ideal to suggest additional analyses to the authors, I strongly recommend further efforts to enhance the article's relevance.

Major issues

I suggest revising the titles of all figures and tables to include details about the population group, location, year of data collection, and sample size (n = ).

Lines 133-134: I understand that "nutrition practices" refer to the overall consumption of selected healthy and unhealthy eating markers, not specifically when purchased in school vicinities. However, there is a potential for misunderstanding when the authors present their conclusions (lines 324-327). To avoid confusion, despite the potential redundancy, I recommend that the authors explicitly clarify this in the methodology and throughout the text. They can use terms such as "general healthy eating markers" and emphasize that the information pertains to the last 30 days.

Lines 131-137: Please, clarify whether the translation and pre-test of the Global School Health Survey (GSHS) questionnaire were part of the referenced validation study [20] or if these steps were conducted by the research team during the planning phase of this research.

Lines 141-142: The authors should provide a general statement elucidating the conceptualization of healthiness adopted in the school canteen policy. For instance, specifying criteria such as energy-dense foods, items with added sugar, high-fat foods, and ultra-processed foods would enhance clarity.

Lines 142-143: In my view, stating that "Food environment was defined as all the outlets selling food within 300 meters from the school" may be misleading. The term "food environment" is a construct with established definitions in the literature. It seems you intended to convey that they defined the school vicinities within this radius, which would be a more accurate description.

Lines 142-146: Please provide additional details to ensure that readers fully comprehend how you determined the 300m radius. Did you select a subset of students to investigate the average distance between their homes and the school? If so, could you elaborate on the methodology employed?

Lines 156-157: Clarify the variables to which you are referring in the statement "...association between the variables and the dietary choices...". Specify the particular variables involved in this analysis.

Lines 205 and 208: Upon reviewing your methods, it appears that you did not inquire students about specific foods they purchase in the school neighborhood. However, the term "snack" is used here. In the methods section, clearly specify the nature of the questions asked, and subsequently, revise the results and discussion sections to eliminate any ambiguities. For instance, in line 208, where you report results about dietary trends, it seems to refer to the consumption of eating markers in the last 30 days, regardless of the place of purchase and consumption. However, in Table 5, the outcome variable is labeled "Consuming energy-dense food from outlets," causing confusion again.

Lines 201-212: I recommend moving these individual level results to line 192, right after Table 3 (including the Figure 2). Then you bring the food environment results and, in the last place, the associations between them.

Table 5: Since you present odds ratios in your results, I assume logistic regression analyses were performed instead of chi-squared tests. However, this is not explicitly mentioned in your methods. Additionally, it might be more accurate to label the variable you named as "Availability of unhealthy food" as "Presence of establishments selling any unhealthy food."

If you possess information on the specific types of foods available for purchase in the food establishments and details about the consumption of these items by the students, I strongly recommend conducting further analyses to demonstrate these associations. This would provide a more nuanced understanding beyond the association between the presence of establishments selling ANY unhealthy food and the consumption of ANY unhealthy food.

Table 6: It seems that you are referring to the consumption of ANY unhealthy food (yes vs. no). Readers might find it particularly valuable to observe relationships stratified by the types of food items available for purchase and those actually consumed. Moreover, I am unclear about the meaning of "Coefficient 95% CI." Where is the 95% confidence interval, and what does this coefficient represent? Is it a beta? You mentioned Spearman correlation analysis in the methods, but it seems incongruent in the context of these two variables.

Furthermore, both association analyses presented in Tables 5 and 6 should be multivariable analyses instead of just bivariate analyses. There are potentially significant confounding variables at both the individual and school neighborhood levels (especially socioeconomic status), and it would be essential to adjust the analyses accordingly.

Minor issues

Lines 73-74: Is the information about school types relevant in the context of the study? It seems to me that it doesn’t add information.

Line 97: I don’t understand what is “economic status of the study”.

I recommend formatting the tables according to international standardized patterns. Additionally, I suggest combining the information from Tables 1 and 2 into a single table, making the necessary changes in its title. Please ensure the titles of Figure 1 are included, and correct the numbering of Figure 2, which is currently labeled as 1.

Regarding Figure 2, showing snacks consumed by students, it does not seem to be referenced in the results section. Please address this inconsistency.

For Table 4, include the results for the role of health and unhealthy food categories. Furthermore, add information about the statistical tests performed in the footnotes of the tables.

In the results section, avoid using statements such as "As presented in Table 2...". Instead, simply reference the titles of the tables and figures within parentheses.

A major language review is necessary to enhance readability throughout the manuscript.

Obs: Even though I downloaded the figures, I couldn't read information from the maps due to the very low resolution, making it difficult to visualize. I’m asking the editor for a solution to this problem.

6. PLOS authors have the option to publish the peer review history of their article (what does this mean?). If published, this will include your full peer review and any attached files.

Reviewer #1: No

Reviewer #2: **Yes: **Laís Vargas Botelho

---

## [Author Response · Author response to Decision Letter 0]

8 Feb 2024

Dear Sir/Madam 

I would like to express my sincere gratitude for the time you dedicated to reviewing my manuscript. Your insightful comments have significantly contributed to enhancing the quality of my research output. I have diligently addressed each of your comments, incorporating comprehensive information to strengthen the validity of the content. Additionally, I have included relevant information that aligns with and supports your valuable feedback. 

Thank you !

---

## [Decision Letter · Decision Letter 1]

21 May 2024

PONE-D-23-28601R1Exploring the relationship between the food environment and preferences among schoolchildren in a low socioeconomic community in Sri Lanka: A GIS-based Assessment.PLOS ONE

Dear Dr. Senevirathne,

Thank you for submitting your manuscript to PLOS ONE. After careful consideration, we feel that it has merit but does not fully meet PLOS ONE’s publication criteria as it currently stands. Therefore, we invite you to submit a revised version of the manuscript that addresses the points raised during the review process.

We look forward to receiving your revised manuscript.

Kind regards,

Larissa Loures Mendes, Ph.D.

Academic Editor

PLOS ONE

Journal Requirements:

Reviewers' comments:

Reviewer's Responses to Questions

**Comments to the Author**

1. If the authors have adequately addressed your comments raised in a previous round of review and you feel that this manuscript is now acceptable for publication, you may indicate that here to bypass the “Comments to the Author” section, enter your conflict of interest statement in the “Confidential to Editor” section, and submit your "Accept" recommendation.

Reviewer #3: All comments have been addressed

2. Is the manuscript technically sound, and do the data support the conclusions?

Reviewer #3: Yes

3. Has the statistical analysis been performed appropriately and rigorously? 

Reviewer #3: Yes

4. Have the authors made all data underlying the findings in their manuscript fully available?

Reviewer #3: Yes

5. Is the manuscript presented in an intelligible fashion and written in standard English?

Reviewer #3: Yes

6. Review Comments to the Author

**Reviewer #3:** The article is well-structured, and the authors have addressed all the recommendations from the reviewers. I have only a few minor revisions to suggest for improving the article.

In the methods (Line 97), it is important to provide more information about the poverty indicator, including the data source - Department of Census - and a description of the variables included in this indicator.

The authors have presented frames instead of tables. I recommend revising the presentation and using the format of a table rather than a frame.

Although the maps have significantly improved in quality compared to the previous version, it is still difficult to see the points. I recommend increasing the size of the points on the maps to enhance visibility.

7. PLOS authors have the option to publish the peer review history of their article (what does this mean?). If published, this will include your full peer review and any attached files.

Reviewer #3: **Yes: **Nayhanne Gomes Cordeiro

---

## [Author Response · Author response to Decision Letter 1]

4 Jun 2024

I appreciate the time and effort invested by the reviewers and the editorial team in evaluating my manuscript titled “Exploring the Relationship between Food Environment and Preferences of School Children in a Low-Socioeconomic Community in Sri Lanka: A GIS-based Assessment. I am grateful for the constructive feedback provided, and I have carefully considered each comment in the revision of my manuscript. We found that each comment provided by both reviewers is highly valuable, and contributes to ensuring the quality of the manuscript. 

The summary of the reviewer’s comments and clear and concise responses to the reviewer's comments have been addressed, and specific changes are incorporated into the revised manuscript.

Therefore, I assure you that the second revised version of the manuscript fully adheres to the journal’s guidelines and policies. 

Thank you for considering my manuscript for publication in PLOS ONE. I appreciate your time and consideration.

Sincerely,

---

## [Editor Report · Decision Letter 2]

26 Jun 2024

Exploring the relationship between the food environment and preferences among schoolchildren in a low socioeconomic community in Sri Lanka: A GIS-based Assessment.

PONE-D-23-28601R2

Dear Dr. Senevirathne,

We’re pleased to inform you that your manuscript has been judged scientifically suitable for publication and will be formally accepted for publication once it meets all outstanding technical requirements.

Kind regards,

Larissa Loures Mendes, Ph.D.

Academic Editor

PLOS ONE

---

## [Editor Report · Acceptance letter]

4 Jul 2024

PONE-D-23-28601R2 

PLOS ONE

Dear Dr. Senevirathne, 

I'm pleased to inform you that your manuscript has been deemed suitable for publication in PLOS ONE. Congratulations! Your manuscript is now being handed over to our production team.

Kind regards, 

on behalf of

Dr. Larissa Loures Mendes 

Academic Editor

PLOS ONE